# Oral Fluid Concentrations and Pharmacological Effects of Clephedrone and Methylone in Humans

**DOI:** 10.3390/ijms27010089

**Published:** 2025-12-21

**Authors:** Lourdes Poyatos, Melani Núñez-Montero, Olga Hladun, Georgina De la Rosa, Soraya Martín, Sebastian Videla, Silvia Martínez-Couselo, Mireia Ventura, Nunzia La Maida, Annagiulia Di Trana, Francesco Paolo Busardò, Marta Torrens, Simona Pichini, Clara Pérez-Mañá, Magí Farré, Esther Papaseit

**Affiliations:** 1Department of Clinical Pharmacology and Biochemistry, Hospital Universitari Germans Trias i Pujol and Institut de Recerca Germans Trias i Pujol (HUGTiP-IGTP), Carretera de Canyet S/N, 08916 Badalona, Spain; lpoyatos@igtp.cat (L.P.); nmmelani@gmail.com (M.N.-M.); ohladun.germanstrias@gencat.cat (O.H.); grosalo.germanstrias@gencat.cat (G.D.l.R.); smartins.mn.ics@gencat.cat (S.M.); svidelac.germanstrias@gencat.cat (S.V.); smartinezcou.germanstrias@gencat.cat (S.M.-C.); cperezm.mn.ics@gencat.cat (C.P.-M.); epapaseit@researchmar.net (E.P.); 2Department of Pharmacology, Therapeutics and Toxicology, Universitat Autònoma de Barcelona (UAB), 08193 Cerdanyola del Vallés, Spain; 3Energy Control, Associació Benestar i Desenvolupament, Carrer de la Independència 384, 08041 Barcelona, Spain; mireia@energycontrol.org; 4National Center On Addiction and Doping, National Institute of Health, Viale Regina Elena 299, 00161 Rome, Italy; nunzia.lamaida@iss.it (N.L.M.); annagiulia.ditrana@iss.it (A.D.T.); simona.pichini@iss.it (S.P.); 5Department of Excellence-Biomedical Sciences and Public Health, Università Politecnica delle Marche, 60121 Ancona, Italy; f.busardo@univpm.it; 6Drug Addiction Unit, Parc de Salut Mar, Hospital del Mar Research Institute, 08003 Barcelona, Spain; mtorrens@researchmar.net

**Keywords:** pharmacology, new psychoactive substances, synthetic cathinones, clephedrone, 4-chloromethcathinone, 4-CMC, methylone, subjective effects

## Abstract

Synthetic cathinones represent the second most frequently reported group of new psychoactive substances identified annually, according to the United Nations. It remains unknown whether specific derivatives differ in the onset of effects related to absorption kinetics. Clephedrone (4-chloromethcathinone, 4-CMC) has been among the most frequently seized cathinones in recent years; however, available data on its pharmacology and abuse potential remain scarce. A non-controlled, prospective, observational study was conducted involving eight healthy volunteers (six women) who self-administered a single oral dose of clephedrone (100 or 150 mg). Study variables were assessed at baseline and over a 5-h period following administration, including vital signs and subjective effects. Oral fluid concentrations of clephedrone and cortisol were determined. For comparison, this article also presents previously unpublished data from a pilot study in which 12 healthy male participants received 150 or 200 mg of methylone under comparable conditions to evaluate effects. Results indicated that both clephedrone and methylone produced stimulant-like subjective effects. However, clephedrone exhibited a delayed onset and peak of effects compared with methylone, indicating a clinically relevant pharmacokinetic difference. Both substances were detected in oral fluid, with peak concentrations occurring later following clephedrone administration, consistent with its delayed pharmacodynamic profile.

## 1. Introduction

Synthetic cathinones are a class of laboratory-produced psychoactive substances chemically related to cathinone, a natural stimulant found in the khat plant (Catha edulis). This group represents one of the most significant categories among new psychoactive substances (NPS), ranking as the second largest group of newly identified substances each year according to the United Nations Office on Drugs and Crime (UNODC), surpassed only by synthetic cannabinoids [1].

First-generation synthetic cathinones are those that emerged in the late 2000s and early 2010s as non-controlled alternatives to traditional illicit stimulant drugs. In this generation, some of the most commonly consumed included mephedrone (4-methylmethcathinone), methylone (3,4-methylenedioxy-N-methylcathinone), methylenedioxypyrovalerone (MDPV), alpha-pyrrolidinopentiophenone (α-PVP), butylone (β-keto-N-methylbenzodioxolylbutanamine), and pentedrone (α-methylaminovalerophenone). Due to their harmful effects, abuse potential, and risks to public health, these substances were promptly banned [2,3]. These compounds demonstrate variability in pharmacological potency and effects, metabolic processes, associated risks, and acute toxicity in both laboratory animals and humans [4,5,6,7,8,9,10,11,12,13,14,15,16].

Because of these regulatory measures, a second generation of synthetic cathinones emerged in the illicit market. These substances were designed to circumvent existing regulations by slightly modifying the chemical structure while maintaining similar pharmacological effects. Some cathinones included in this generation are clephedrone (4-chloromethcathinone, 4-CMC), n-ethyl-pentedrone (NEP, α-EAPP), 3-methylmethcathinone (3-MMC), 4-methylethcathinone (4-MEC), or ethylhexedrone (NEH, HEXEN). These compounds exhibit variability in potency, pharmacological effects, metabolic pathways, and associated risks [17,18,19,20,21].

According to the European Monitoring Centre for Drugs and Drug Addiction (EMCDDA), synthetic cathinones constituted approximately 50% of the total quantity of new psychoactive substances seized by EU Member States in 2021. In fact, the synthetic cathinones most reported were 3-CMC (34%), 3-MMC (8%), clephedrone (6%), and N-ethylhexedrone (2%) [22].

One clinically relevant unanswered question is whether first- and second-generation synthetic cathinones differ in their pharmacological effects and time course, particularly in relation to pharmacokinetic parameters such as time to reach maximal plasma concentration (T_max_) and duration of effects relative to elimination half-life. Some data exist for first-generation compounds such as mephedrone and methylone, which show a rapid onset and relatively fast elimination. However, information on second-generation cathinones remains scarce. At present, no published data exists regarding the pharmacological effects or pharmacokinetics of clephedrone in humans.

Clephedrone was first characterized in 2014 after its appearance on the online drug market [23]. As in the case of several other NPS, current knowledge of clephedrone’s pharmacology, toxicology, and pharmacokinetics remains limited; thus, existing information on its acute and adverse effects is largely based on anecdotal user accounts and documented intoxication cases. For instance, in 2017, a systematic search was carried out to collect self-reported experiences from various synthetic cathinone users, including clephedrone, published on internet fora [24]. Clephedrone users reported consuming doses ranging from 50 to 1000 mg, depending on their tolerance and prior experience with other cathinones. They described effects lasting between 2 and 4 h, with an 3,4-Methylenedioxymethamphetamine (MDMA)-like profile, characterized by euphoria, increased energy, sociability, empathy, and sexual desire, accompanied by visual and auditory hallucinations. Users also described some adverse effects related to clephedrone consumption, such as anxiety, apathy, jaw tension, and nystagmus.

In this article, we report the results of an observational study evaluating the effects and oral fluid pharmacokinetics of self-administered oral clephedrone in humans. Since results showed an unexpected, delayed onset of effects in comparison to other synthetic cathinones described in previous publications, it was decided to include, for comparison purposes, unpublished results of a pilot clinical trial of methylone that follows a faster onset of actions and kinetics, as do other synthetic cathinones. Given that methylone was included solely for comparative purposes, this manuscript focuses primarily on clephedrone. Data on methylone—including its epidemiology, pharmacological effects, pharmacokinetics, metabolism, and acute toxicity—have already been described in some previous publications [23,25,26]. A comparative analysis of clephedrone and methylone could yield valuable insights into the similarities and differences among synthetic cathinones, considering their molecular differences [25].

Indeed, the primary molecular difference between clephedrone and methylone lies in the substituents attached to their aromatic (phenyl) ring (Figure 1) [23,27].

Clephedrone is a halogenated cathinone derivative, characterized by the presence of a chlorine atom directly attached to the phenyl ring at the 4-position. This chlorine atom increases molecular lipophilicity and exerts an electron-withdrawing effect on the aromatic ring. A structural isomer of clephedrone is 3-chloromethcathinone (3-CMC), which differs solely in the position of its chlorine atom [23,27].

Methylone is the beta-keto analog of 3,4-Methylenedioxymethamphetamine (MDMA, midomafetamine, ecstasy). Its distinctive structural feature is the methylenedioxy ring fused to the benzene ring at the 3 and 4 positions. The methylenedioxy moiety modulates electron distribution, ring conformation, and metabolic pathways typical of methylenedioxyphenethylamines [27,28].

These molecular characteristics can explain the differences and similarities in its pharmacological properties.

In relation to its molecular mechanism of action, clephedrone mainly acts as a substrate for monoamine transporters of dopamine (DAT), serotonin (SERT), and norepinephrine (NET), leading to a reversal of the normal direction of transporter flux [29,30,31]. It seems more active on DAT and NET with much weaker action at SERT. Its monoamine uptake inhibition profile has also been evaluated, but clephedrone demonstrated lower potency as an inhibitor of the reuptake of monoamines. The described molecular profile produces more psychostimulant than empathogen effects [32].

Methylone’s molecular action is related to its action as a substrate for monoamine transporters of dopamine (DAT), serotonin (SERT), and norepinephrine (NET), leading to the release of these monoamines as MDMA and less activity on reuptake inhibition. Its activity is similar in the three transporters, with more action on serotonin than clephedrone. The described molecular profile seems to produce mixed empathogen/psychostimulant effects [32,33].

In a study of drug discrimination in rodents, clephedrone fully substituted for MDMA and displayed more potency at producing MDMA-like discriminative stimulus effects than cocaine- or methamphetamine-like discriminative stimulus effects. Also, this cathinone produced locomotor effects like MDMA with a slow onset and long-acting effects. Methylone presented similar results in drug discrimination studies, fully substituting (>80%) for MDMA in rats [34].

The molecular structures of both synthetic cathinones determine their metabolism. The principal metabolic pathways of clephedrone were determined in urine from this study [35]. This analysis identified three main metabolites: one following a reduction of the beta-keto group to form hydroxy-4-CMC, a N-demethylation to desmethyl-4-CMC (nor-4-CMC), and a reduction followed by demethylation to form hydroxy-nor-4-CMC, and some conjugates. Compared to other synthetic cathinones, clephedrone demonstrated a slow metabolite formation with a long half-life and low intrinsic clearance. Also, a study in vitro expanded this metabolic map by identifying further metabolites through the incubation of the substance in human hepatocytes [17].

Due to its molecular structure, methylone undergoes metabolic transformations characteristic of methylenedioxy-substituted compounds as MDMA. Metabolic pathways included O-demethylenation to 3,4-dihydroxy-N-methylcathinone (HHMC), followed by O-methylation to 4-hydroxy-3-methoxy-N-methylcathinone (HMMC), the most abundant metabolite in urine, and subsequent conjugation. These pathways include the actions of cytochrome P450 2D6 (CYP2D6) and the catechol-O-methyltransferase (COMT). Demethylation of methylone produces the primary amine derivative, 3,4-methylenedioxycathinone (MDC) [3,15].

Based on data, clephedrone was one of the most detected synthetic cathinones in 2015, along with pentedrone, α-PVP, and α-PHP. In that year, clephedrone started to be detected in biological samples, typically in combination with amphetamine, clonazepam, and pentedrone [36]. Indeed, published data about blood concentrations of clephedrone in cases of intoxication confirmed this pattern of polydrug use. In samples collected from 15 forensic cases, beyond the presence of clephedrone, several other compounds were detected, including alcohol, amphetamine, benzodiazepines, MDMA, MDA, 3-MMC, and cocaine [37]. Blood concentrations of clephedrone from samples obtained from traffic controls in non-fatal cases (n = 9) ranged from 1.3 to 75.3 ng/mL (mean of 21.1 ng/mL), whereas in fatal cases, concentrations ranged from 56.2 to 1870 ng/mL (mean of 547 ng/mL). Also, clephedrone has been detected in several cases of intoxication related to other NPS [38,39,40].

Regarding clephedrone’s toxicity, this cathinone has been associated with mitochondrial toxicity through an impairment of the mitochondrial respiratory chain [32]. Moreover, in silico studies aimed to predict the toxicity of clephedrone obtained results that suggested high susceptibility of the gastrointestinal system and the lungs to the toxicity of this cathinone, potential risk of Deoxyribonucleic Acid (DNA) damage and carcinogenicity after prolonged exposure, and notable cardiotoxicity [41].

To date, no study in humans has been published that characterizes the acute effects profile of clephedrone, nor is there information on its concentrations in oral fluid following recent consumption. Accordingly, this study aimed to characterize the pharmacological effects of orally self-administered clephedrone under naturalistic conditions and to relate its time course to corresponding salivary concentration profiles. To compare the pharmacological profile and abuse potential of clephedrone, this article also presents unpublished data from a pilot clinical trial investigating pharmacological effects and oral fluid concentrations of methylone. The inclusion of both studies in the same manuscript—despite their very different designs (an observational study and a randomized clinical trial)—will permit the comparison of the onset-time-course of pharmacological effects, and pharmacokinetics of two synthetic cathinones from different generations.

## 2. Results

All the administered doses of clephedrone and methylone were well tolerated, and no serious adverse events were observed.

### 2.1. Participants

The eight participants (6 females, 2 males) included in the naturalistic clephedrone study had a mean age of 36 ± 8 years (range 30–54), weighed 65 ± 15 kg (range 46–87), with a mean body mass index (BMI) of 22.4 ± 4.8 kg/m^2^ (range 17.0–29.3). Seven participants admitted being smokers, but only 2 of them smoked daily (range 1–5 cigarettes/day), 7 participants were alcohol users (mean of 12.1 ± 6.4 g/day, range 0.5–20), and all of them were MDMA (mean of 5.0 ± 5.7 uses/year, range 0–15) and cannabis users (mean of 40.4 ± 105.2 uses/year, range 0–300). All the subjects tested negative in the urine drug tests performed at the beginning of the session.

The 12 participants included in the controlled methylone study had a mean age of 24 ± 1 years (range 22–24), weighed 70 ± 7 kg (range 62–87), and had a mean BMI of 21.6 ± 3.4 kg/m^2^ (range 18.0–27.9). Five participants reported being daily smokers (mean of 8.8 ± 1.8 cigarettes/day, range 6–10), all of them reported previous use of alcohol (mean of 12.4 ± 4.1 g/day, range 7.0–19.2), MDMA (mean of 6.3 ± 3.2 uses/year, range 3–12) and cannabis (mean of 81.8 ± 100.6 uses/year, range 0–300). All subjects tested negative in the urine drug tests performed at the beginning of the session.

### 2.2. Physiological Effects

Physiological effects of clephedrone and methylone are represented in Figure 2 and summarized in Table 1.

Both clephedrone and methylone produced an increase of non-invasive systolic blood pressure (SBP), diastolic blood pressure (DBP), and heart rate (HR) after a single oral administration, and only the former induced an increase of Tenperature (T). Cardiovascular effects produced by clephedrone had a retarded onset at 2 h after self-administration compared to methylone, which produced fast effects that started at 0.5 h after administration. Compared to baseline, clephedrone significantly increased SBP and DBP from 2 to 4 h after oral administration, HR from 3 to 5 h, and T from 0.5 to 2 h. Clephedrone produced maximum effects (E_max_) of +16.81 mmHg for SBP, +13.6 mmHg for DBP, and +17.19 bpm for HR at 2.5–3.0 h (T_max_). In the case of methylone, significant differences in the time course compared to baseline were found between 0.5 and 2 h for SBP, 1 to 1.5 h for DBP, and 0.5 to 6 h for HR. Also, the maximum effects of methylone on SBP (+27.04 mmHg) and HR (+28.75 bpm) were higher and appeared earlier (T_max_ of 0.75 h) compared to clephedrone. Methylone produced milder effects on DBP (+3.75 mmHg) than clephedrone.

### 2.3. Subjective Effects

Clephedrone and methylone induced significant changes in several subjective effects collected using visual analog scales (VAS), the short form of the Addiction Research Center Inventory (ARCI), and the Evaluation of Subjective Effects of Substances with Abuse Potential questionnaire (VESSPA-SSE) questionnaires (Figure 3 and Table 1).

Overall, both synthetic cathinones induced subjective effects that were reflected as significant changes in VAS related to stimulation (VAS ‘stimulated’, ‘high’), wellbeing (VAS ‘good effects’, ‘liking’), altered perception (‘changes in lights’, ‘changes in hearing’, ‘different or changed body feeling’, ‘different surroundings’), concentration (VAS ‘concentration’), drowsiness (VAS ‘drowsiness’), empathy (‘open’, ‘trust’, ‘feeling close to others’, ‘I want to be with other people’, ‘I want to hug someone’) and sexual drive (VAS ‘sexual desire’, ‘sexual arousal’). Hallucinogen effects were not reported.

Participants who received clephedrone reported high scores for certain subjective effects, with E_max_ values exceeding 40 mm for effects such as ‘intensity’, ‘stimulated’, ‘high’, ‘good effects’, ‘liking’, ‘open’, ‘trust’, ‘feeling close to others’, and others with E_max_ values superior to 30 mm such as ‘different or changed body feeling’, ‘I want to be with other people’, ‘I want to hug someone’. Effects such as ‘concentration’, ‘sexual desire’, and ‘sexual arousal’ also obtained high scores of E_max_ exceeding 20 mm. In most of said effects, statistically significant differences with baseline values were obtained at 2, 3 and 4 h, except for ‘concentration’ (3, 4 h), ‘different body feeling’ (3, 4 h), ‘trust’ (2, 3, 4, 5 h), ‘I want to hug someone’ (4 h) and ‘sexual drive’ (3, 4 h). T_max_ of these subjective effects was 3 h.

For methylone, subjective effects rated with high scores of Emax exceeding 20 mm were ‘intensity’, ‘stimulated’, ‘high’, ‘good effects’, and ‘liking’. Other effects with relevant Emax values superior to 15 mm included ‘open’, ‘trust’, ‘feeling close to others’, ‘I want to be with other people’, and ‘I want to hug someone’. All these subjective effects, plus ‘different or changed body feeling’, had significant differences along their time course compared to baseline values; those related to stimulation, wellbeing, and altered perception at 0.5, 1, 1.5, and those related to empathy at 1 and 1.5 h. T_max_ of these subjective effects was 0.5 and 0.75 h.

Regarding ARCI subscales, both synthetic cathinones obtained the highest E_max_ values in morphine-benzedrine group (MBG) (euphoria), Benzedrine (BG) (intellectual efficiency and energy), and Amphetamine (A) (amphetamine-like effects) subscales. In particular, clephedrone produced significant changes in time course compared to baseline of all the ARCI subscales; these differences were detected from 1 to 5 h for pentobarbital-chlorpromazine-alcohol group (PCAG) (sedation), lysergic acid diethylamide (LSD) (dysphoria), and A, from 2 to 5 h for MBG, and from 1 to 2 h for BG. In the case of methylone, significant differences in time course were obtained at 1 h for MBF and BG, and from 1 to 2 h for A. Clephedrone’s effects collected using ARCI were maximum (T_max_) at 2 and 2.5 h, whereas for methylone, maximum effects appeared earlier at 1 h post-administration.

For VESSPA subcales, participants who self-administered clephedrone reported the highest Emax for psychosomatic anxiety (ANX), pleasure and sociability (SOC), and activity and energy (ACT) subscales. Regarding the time course of these effects, significant changes compared to baseline were detected at 5 h for S, from 3 to 5 h for ANX, from 2 to 4 for SOC and ACT, and from 2 to 3 h for PS. Regarding methylone, only the SOC subscale obtained E_max_ higher than 1 out of a maximum of 4. Significant differences along the time course were detected for ANX from 1 to 4 h, and for SOC and ACT at 1 h post-administration. Moreover, T_max_ of clephedrone ranged from 2 to 3 h, whereas for methylone T_max_ obtained was of 1 h.

### 2.4. Oral Fluid Concentrations

Clephedrone and methylone concentrations time course in oral fluid is represented in Figure 4.

Oral fluid concentrations of clephedrone increased steadily until reaching a mean maximum (C_max_) of 6405.4 ± 3494.8 ng/mL at 3 h (2.0–5.0) (T_max_) after its oral self-administration. The calculated AUC_0–5 h_ was of 10,730.6 ± 3627.0 ng/mL·h. Participants ended the experimental session with remaining concentrations of clephedrone in oral fluid, although concentrations had decreased to at least half of the C_max_ for 7 of the participants.

Regarding methylone, oral fluid concentrations increased rapidly, achieving a C_max_ of 17,246.2 ± 7555.7 ng/mL at 2.0 (1.0–2.0) h post-administration. The AUC_0–6 h_ obtained from methylone’s concentrations was 44,686.2 ± 24,156.3 ng/mL·h. After reaching their peak, oral fluid concentrations started to rapidly decrease compared to clephedrone. Although concentrations were still significant at 6 h, they were 11 times lower than C_max_ at the time participants ended the experimental sessions.

Concentrations of cortisol in oral fluid after clephedrone are shown in Figure 5. Cortisol oral fluid concentrations increased following clephedrone administration. Mean peak concentrations (T_max_) were observed at 3 h (1.0–4.0). Concentration increases were statistically significant versus baseline at 3 and 4 h after administration (Dunnett’s test).

## 3. Discussion

To the best of our knowledge, this is the first study aimed at assessing the acute pharmacological effects and oral fluid concentrations of the synthetic cathinone clephedrone orally self-administered by recreational users in a naturalistic setting. Moreover, this manuscript presents results from a controlled study that evaluated acute effects and oral fluid concentrations of methylone, a synthetic cathinone with a similar profile to mephedrone and MDMA.

The main finding is that, after oral self-administration, clephedrone induced a plethora of pharmacological effects resembling those described for methylone, characterized by increased blood pressure and heart rate, feelings of wellbeing, stimulation, euphoria, and heightened sociability. However, the acute effects of clephedrone had a delayed onset and peak compared to methylone. This difference in time course was also observed in the results of their oral fluid concentrations, which reflected a later peak in the case of clephedrone concentrations and cortisol response.

As expected, according to users’ reports, clephedrone exhibited a similar profile to MDMA, including sensations of euphoria, increased energy, and pro-social effects. Although previous reports described visual and auditory hallucinations after clephedrone use, our results did not evidence any hallucinogenic effects. In terms of physiological effects, clephedrone produced a lower increase of systolic blood pressure than methylone and MDMA or cathinone [14,42,43]. It is noteworthy that peak effects on SBP of clephedrone appeared at 3 h, whereas methylone produced faster effects that were maximum at 0.75 h and returned to baseline at 3 h. Heart rate was also significantly affected by both synthetic cathinones, with later effects after clephedrone administration that peaked at 3 h. Effects on heart rate were not normalized 5 or 6 h after administration, in fact, remaining constant at approximately 20 bpm above baseline. For methylone, it has been demonstrated to cause significant and prolonged increases in heart rate in both naturalistic and controlled environments that persisted for 8–10 h [14,44]. In terms of subjective effects, clephedrone produced effects similar to methylone, but with a delayed onset and peak. Under similar naturalistic conditions, clephedrone also exhibited a delayed time course compared to methylone, which reached peak effects 2 h following self-administration [44]. Clephedrone induced higher subjective effects at lower doses than methylone, suggesting greater potency; however, the lack of placebo expectancy could also have influenced the naturalistic study. Overall, these results position clephedrone as a psychostimulant with an abuse potential comparable to prototypical MDMA, alongside substances like natural cathinone, methylone, mephedrone, and 3-MMC [14,25,42,43,45,46,47,48]. For instance, previous studies about mephedrone described stimulant effects accompanied by altered perception, feelings of wellbeing and increased empathy that peaked at 0.75 h and returned to baseline after 2–3 h [13,48,49]. Similarly, when orally administered, cathinone induces activation and euphoria [42,43]. The increase in cortisol concentrations after clephedrone has been observed after the administration of other psychostimulants (cocaine and amphetamine)/MDMA in plasma/oral fluid [50,51,52,53]. Thus, clephedrone produces pharmacological effects similar to other orally consumed synthetic cathinones, like methylone and mephedrone –both of which are comparable to MDMA–, but with a delayed onset.

The time course of clephedrone pharmacological effects on physiological/subjective effects and its influence on cortisol concentrations relates to the results obtained from its oral fluid concentrations, which indicate a slower pharmacokinetics compared to methylone, with peak concentrations observed at 3 h. This could explain the differences with other synthetic cathinones that exhibit faster kinetics, such as mephedrone, with T_max_ values of 1.25 h in plasma and 2 h in oral fluid [13,49], and methylone, with T_max_ values described at 2 h in plasma and oral fluid [15,44,54].

Regarding limitations, the inclusion of two studies with distinct designs (an observational study and a randomized clinical trial) is a relevant point that must be acknowledged. However, given the highly interesting results observed following clephedrone self-administration, we believe that a direct comparison with methylone provides original and previously unobserved differences. Two different synthetic cathinones can induce similar overall pharmacological effects, yet present markedly different profiles in terms of time of onset and time course. This finding is of significant clinical relevance. Therefore, the inclusion of both studies is essential for understanding that molecules within the same class can exhibit distinct pharmacokinetic profiles.

For the controlled methylone study, the main limitations were the exclusion of female participants, the limited number of subjects, and the restricted range of doses included. Although the sample size was sufficient to achieve statistically significant results regarding the time course, a larger sample size might have revealed differences between the two doses, which were ultimately aggregated due to the lack of observed effect differences. Finally, cortisol concentration data were unavailable for comparison.

In particular, the study of clephedrone had limitations inherent to its design as an observational–naturalistic study. The sample size used in the naturalistic study was insufficient to explore sex influence in the acute effects and/or pharmacokinetics of clephedrone. Moreover, differences in effects based on the doses could not be detected, nor could a dose-response relationship be established. The study was open-label, so it is susceptible to an expectancy bias. However, as explained in other studies, the observational approach and the results presented in the present paper can produce adequate results when controlled clinical trials cannot be performed [14,16]. Due to the delayed onset of clephedrone effects, additional assessments after 5 h would have allowed us to define the time course of its pharmacological effects and oral fluid concentrations until returning to baseline.

Another limitation, as previously commented, is that a formal comparative analysis between the two cathinones was not possible due to the significant difference in study design (an observational study versus a randomized clinical trial). This disparity in design is particularly relevant because it could have directly influenced the reported subjective effects. Specifically, it is generally accepted that open-label studies often elicit more pronounced subjective effects than controlled, double-blind studies.

In this study, the molecular different aromatic substitutions in the clephedrone and methylone structures and supposed different actions on monoamine release did not produce differences in the pharmacological effects, except that the intensity of subjective effects was higher for clephedrone, probably related to the open administration of the substance and/or expectancy. The molecular differences could be responsible for changes in the absorption, distribution, and elimination of the substances, which result in a delayed onset of effects and pharmacokinetics.

## 4. Materials and Methods

### 4.1. Study Design

Two studies with different designs were carried out to evaluate the pharmacological effects of clephedrone and methylone in humans after their oral administration.

The study evaluating the acute effects of clephedrone was designed as a non-controlled, prospective, observational, and naturalistic study. The procedures and evaluations included in this study have been described in previous studies evaluating other synthetic cathinones (mephedrone, methylone, alpha-PVP, NEP, HEXEN), phenethylamines (2C-B and 2C-E), or synthetic cannabinoids (JWH-122, JWH-210, UR-144) [14,16,49,55,56,57,58,59,60].

This study consisted of one experimental session in which participants (N = 8; 6 women, 2 men) selected their oral dose of clephedrone based on their drug use experience and preferences. According to their dose selection, 3 women and 1 man ingested 100 mg of clephedrone, while the remaining 3 women and 1 man opted for a higher dose of 150 mg. Hence, the mean dose of self-administered clephedrone in this study was 125 mg. Participants were recruited by word of mouth by the harm reduction organization Energy Control in Barcelona (Associació Benestar i Desenvolupament, ABD).

Methylone pharmacology was evaluated in a study designed as a randomized, double-blind, placebo-controlled, crossover clinical trial. Four pilot studies were conducted administering methylone at oral doses of 50 (n = 3), 100 (n = 6), 150 (n = 5), and 200 mg (n = 7) [15]. Participants attended 3 experimental sessions separated by a one-week wash-out period and received two doses of methylone (one dose per experimental session) compared to placebo, except in the last pilot study in which participants received one dose of methylone, compared to MDMA and placebo. However, in this manuscript, only the results corresponding to the highest methylone doses (n = 12; 150 and 200 mg; mean dose = 179.2 mg) are presented, to enable comparison with clephedrone self-administration. Participants were recruited by word of mouth in the volunteer database of the clinical pharmacology unit.

Both in the study with clephedrone and in that with methylone, it was required that participants had previous experience with synthetic cathinones, amphetamines, ecstasy, and/or cocaine, free of serious adverse reactions at least six times in their lifetime for the naturalistic study and at least twice in the last year and six times in their lifetime for the controlled study. However, subjects were not eligible if they had any ongoing or recent (within 3 months before inclusion) organic illness, had undergone major surgery, or had history of mental disorders, including substance use disorder (except for nicotine), as defined by the Diagnostic and Statistical Manual of Mental Disorders-5.

All participants received economic compensation for their participation in each study.

The studies were conducted in accordance with the Declaration of Helsinki and approved by the Institutional Review Board (local Human Research Ethics Committee [CEIm Hospital Germans Trias i Pujol, codes PI-18-267 and PI-19-082]). Both studies were conducted following Spanish legislation regarding clinical research (Biomedical Research Law 14/2007). The controlled study evaluating methylone, as a clinical trial, is registered at the database ClinicalTrials.gov (NCT05488171).

### 4.2. Procedures

All participants underwent a medical examination and a psychiatric evaluation before enrollment; in the case of the controlled methylone study, this examination also included an electrocardiogram (ECG) and blood and urine tests. Additionally, participants were trained in the study procedures and questionnaires to be used during the experimental sessions.

Participants of the naturalistic clephedrone study were summoned at 15:00 h at the private club where sessions were carried out. The club remained closed to the public until the end of the session, approximately at 21:00 h. As a naturalistic study, participants were left free to talk, read, play games, or listen to music in the time between study evaluations. To avoid possible influences between participants, they were asked to refrain from talking about the effects of the self-administered substance. The experimental sessions of the controlled pilot trial were carried out at the Clinical Research Unit (UPIC) of the Germans Trias i Pujol hospital, where participants were summoned at 7:45 a.m. after fasting overnight and remained there for approximately 11 h.

Participants were required to abstain from illicit drug use for two days before the naturalistic study and seven days before the controlled study. Upon arrival, urine samples were collected and analyzed using the Drug-Screen Multi 10TD Test (Multi-Line; Nal Von Minden, Moers, Germany) to verify the absence of drugs of abuse. (amphetamine, barbiturate, benzodiazepine, cocaine, MDMA, methamphetamine, morphine, methadone, tricyclic antidepressants, and tetrahydrocannabinol). Moreover, participants were requested to refrain from pre-session use of alcohol (24 h for the naturalistic study and 48 h for the controlled study) and caffeine/xanthines (24 h). A pregnancy test was performed in participating women using the Clip Test Plus HCG card^®^ (Biosynex Swiss, Delémont, Switzerland).

### 4.3. Drugs

In the naturalistic study, participants brought their own sample of clephedrone obtained by their own means, but the purity of the compound was tested by the harm reduction organization Energy Control (ABD), which offers drug users a drug checking service. Gas chromatography coupled with mass spectrometry (GC/MS) technique was used to analyze the pills of clephedrone and detect possible presence of other common drugs of abuse, such as cocaine, amphetamine and methamphetamine, MDMA, heroin, LSD, and multiple NPS (synthetic cathinones, synthetic cannabinoids, tryptamines, among others). The testing of the pills proved clephedrone purity higher than 95%, as well as the absence of adulterants or toxic components [21].

Methylone hydrochloride of high purity in powder form was acquired from LGC Standards (Teddington, UK) for the controlled study. The Pharmacy Department of the Hospital Universitari Germans Trias i Pujol was responsible for manufacturing, storing, and dispensing identical and opaque soft gelatin capsules containing 50 mg of methylone or a placebo (maltodextrin). Treatment consisted of 5 capsules that combined methylone and/or a placebo to reach the target dose. Details can be found in the previous article [15].

### 4.4. Physiological Effects

In the naturalistic clephedrone study, non-invasive systolic blood pressure (SBP), diastolic blood pressure (DBP), and heart rate (HR) were assessed, while subjects remained seated with an automatic Omron monitor (Omron, Healthcare, Kyoto, Japan) at baseline, 0.50, 1, 1.5, 2, 3, 4 and 5 h (h) after self-administration. Forehead cutaneous temperature was determined at the same time points using a cutaneous infrared thermometer.

In the controlled methylone study, vital signs monitors (Philips Sure Signs VM4 monitors, Philips, Amsterdam, The Netherlands) were used to measure SBP, DBP, HR, and oral temperature at baseline (−30 and −15 min) and 0.25, 0.50, 0.75, 1.5, 2, 3, 4, 6, 8, 10, and 24 h after administration. For these evaluations, participants rested in hospital beds in a position of back elevation. Furthermore, electrocardiograms were constantly monitored throughout the sessions for safety reasons.

### 4.5. Subjective Effects

Subjective effects were evaluated with visual analog scales (VAS), the short form of the Addiction Research Center Inventory (ARCI), and the Evaluation of Subjective Effects of Substances with Abuse Potential questionnaire (VESSPA-SSE). In addition, the Sensitivity to Drug Reinforcement Questionnaire (SDRQ) and a pharmacological identification class questionnaire were used in the controlled study administering methylone.

VAS encompassed several adjectives that participants had to rate from “not at all” (0 mm) to “extremely” (100 mm) according to their sensations. This scale included 28 items: intensity, stimulated, high, good effects, bad effects, liking, changes in distances, changes in colors, changes in shapes, changes in lights, hallucinations (seeing lights or spots), hallucinations (seeing things, animals, insects, or people), changes in hearing, hallucinations (hearing sounds or voices), drowsiness, concentration, dizziness, confusion, different or changed body feeling, unreal body feeling, different surroundings, unreal surroundings, open, trust, feeling close to others, I want to be with other people, I want to hug someone, sexual desire, and sexual arousal [14,44,60].

The validated ARCI 49-item short form was used to evaluate and discriminate the subjective following five subscales that correspond to different classes of drugs of abuse: pentobarbital-chlorpromazine-alcohol group (PCAG) is susceptible to sedative effects, morphine-benzedrine group (MBG) is susceptible to euphoric effects, lysergic acid diethylamide (LSD) is susceptible to dysphoria and somatic symptoms, benzedrine (BG) is susceptible to intellectual efficiency and energy, and amphetamine (A) is susceptible to amphetamine-like effects [14,61].

The VESSPA-SSE questionnaire, used to evaluate the subjective effects of stimulant drugs, The questionnaire includes 36-items and consists of six subscales that measure sedation (S), psychosomatic anxiety (ANX), changes in perception (CP), pleasure and sociability (SOC), activity and energy (ACT), and psychotic symptoms (PS) [14,62].

Moreover, participants enrolled in the controlled methylone study administering methylone, completed two additional questionnaires, the SDRQ [31] and the pharmacological identification class questionnaire. In the SDRQ, participants rated from 1 to 5 “How pleasant was the substance”, an item related to drug liking, and “How much you wanted to use it in that moment”, related to drug wanting [14].

Participants included in the naturalistic clephedrone study completed VAS at baseline, 0.50, 1, 1.5, 2, 3, 4, and 5 h, and ARCI and VESSPA at baseline, 1, 2, 3, 4, and 5 h after self-administering clephedrone. In the controlled study, VAS was performed at baseline and 0.25, 0.50, 0.75, 1, 1.5, 2, 3, 4, 6, 8, 10, and 24 h, and ARCI and VESSPA at baseline and 1, 2, 3, 4, 6, 8, and 10 h after administering methylone. Participants completed SDRQ and VAS sexual desire and arousal at baseline, 1 and 10 h, and the pharmacological class identification questionnaire at 8 h [14,63].

### 4.6. Oral Fluid Concentrations

In both studies, oral fluid samples for the quantification of clephedrone and methylone concentration were collected using Salivette standard tubes. For clephedrone, oral fluid samples were collected at baseline, 0.50, 1, 1.5, 2, 3, 4, and 5 h, whereas for methylone, they were collected at baseline, 0.25, 0.50, 0.75, 1, 1.5, 2, 3, 4, 6, 8, and 10 h. After collection, all samples were centrifuged and stored frozen at −20 °C until analysis. Clephedrone oral fluid concentrations were determined using a previously validated gas chromatography coupled to mass spectrometry systems (GC-MS/MS) [63]. Methylone was quantified via liquid chromatography–tandem mass spectrometry (LC–MS/MS) [54,64,65].

Cortisol oral fluid concentrations were analyzed for clephedrone at the same hours using the electrochemiluminescence immunoassay (ECLIA) method (Elecsys Cortisol II), performed on the Cobas e analyzer platform (Roche Diagnostics GmbH, Mannheim, Germany). A relation between oral fluid concentrations of cortisol and clephedrone could indicate a relationship between pharmacokinetics and effects. Cortisol levels after methylone were not determined.

### 4.7. Statistical Analysis

For the naturalistic clephedrone study, as previously described, subjects were recruited through word of mouth by the ONG Energy Control. Considering the design and requirements of the study, a sample size of 8 individuals was considered optimal to obtain representative results about the pharmacological effects of the substance of interest. For the controlled pilot study of methylone, a minimum of 3 participants per cohort (dose) was considered enough to evaluate doses and select the one for the definitive study [14].

For statistical analysis, values of physiological (SBP, DBP, HR, and T) and subjective effects (VAS, ARCI, and VESSPA-SSE) were baseline–adjusted. Maximum effects (E_max_) and the time needed to reach maximum effects (T_max_) were determined, and the area under the curve up to 5 h (AUC_0–5h_) for clephedrone and up to 6 h (AUC_0–6h_) for methylone was calculated using the trapezoidal rule. E_max_ and AUC values are presented as mean ± standard deviation, whereas T_max_ is presented in median (range).

In both studies, participants did not receive the same doses of clephedrone or methylone, so it was important to determine whether the difference in dose had any impact on the acute effects of these two synthetic cathinones. Furthermore, sex could also have some influence on the results of the naturalistic study. Therefore, a two-way ANOVA with dose and sex as factors was conducted to examine their influence on the clephedrone’s results. Out of all variables analyzed, only 2 out of 87 (2.3%) variables resulted in significance. For methylone, a one-way ANOVA with dose as a factor was conducted, but only 1 of the 87 (1.1%) variables resulted in significance (sex was not considered because all participants were men). These results allowed us to group all participants as just one group of clephedrone and another of methylone, independently of dose or sex, given that neither of those two factors showed a significant influence on the acute effects.

To find possible significant changes through the time course compared to baseline, a Dunnett multiple comparison test was carried out to compare each time point with baseline in both drug conditions. For clephedrone the comparisons were 0–0.5 h, 0–1 h, 0–1.5 h, 0–2 h, 0–3 h, 0–4 h and 0–5 h, and for methylone this analysis included 0–0.25 h, 0–0.5 h, 0–0.75 h, 0–1 h, 0–1.5 h, 0–2 h, 0–3 h, 0–4 h and 0–6 h. Given their different study designs, no comparative analysis between the time courses of both synthetic cathinones was conducted.

For oral fluid concentrations of clephedrone and methylone, only a descriptive analysis was performed to obtain the main pharmacokinetics data, including the maximum concentration (C_max_), the time required to reach maximum concentrations (T_max_), and AUC up to 5 h (AUC_0–5h_) for clephedrone and up to 6 h (AUC_0–6h_) for methylone. These parameters were calculated using the Pharmacokinetic Functions for Microsoft Excel (Joel Usansky, Atul Desai, and Diane Tang-Liu, Department of Pharmacokinetics and Drug Metabolism, Allergan, Irvine, CA, USA). Oral cortisol concentrations after clephedrone were analyzed using Dunnett’s multiple comparison test (each time point with baseline).

Statistical analysis was performed using the software PASW Statistics version 18 (SPSS Inc., Chicago, IL, USA). Differences with *p* values < 0.05 were considered statistically significant.

## 5. Conclusions

This naturalistic study serves as an initial preliminary exploration into defining the acute pharmacological effects and oral fluid concentrations of clephedrone in humans. These findings suggest that the pharmacological effects of clephedrone and methylone, including euphoria, stimulation, and increased sociability, were very similar to those typically associated with psychostimulants such as amphetamines, MDMA, and other cathinones like mephedrone. Peak pharmacological effects of clephedrone appear at 3 h, whereas effects of methylone and mephedrone appear at 0.75–1 h. This is consistent with delayed peak concentrations of clephedrone in oral fluid at 3 h post-administration. Oral fluid could be a suitable non-invasive matrix to detect acute clephedrone and methylone use. When orally administered, synthetic cathinones of different generations can produce an onset and peak effects that differ in the time of appearance depending on their pharmacokinetics parameters. More slow absorption and a delayed time of peak concentrations produce a delayed onset of effects in clephedrone in comparison to methylone. These differences could be relevant for the clinical management of intoxications, alerting that some cathinones derivatives produce rapid effects, whereas others have a later onset.

## Figures and Tables

**Figure 1 ijms-27-00089-f001:**
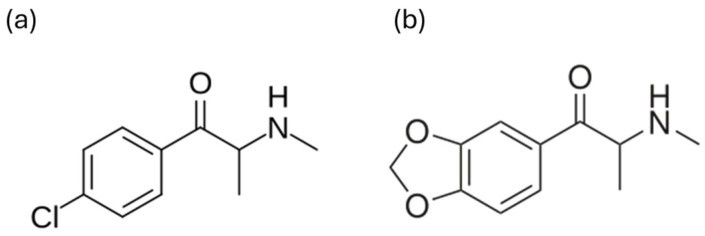
Chemical structures of clephedrone (**a**) and methylone (**b**).

**Figure 2 ijms-27-00089-f002:**
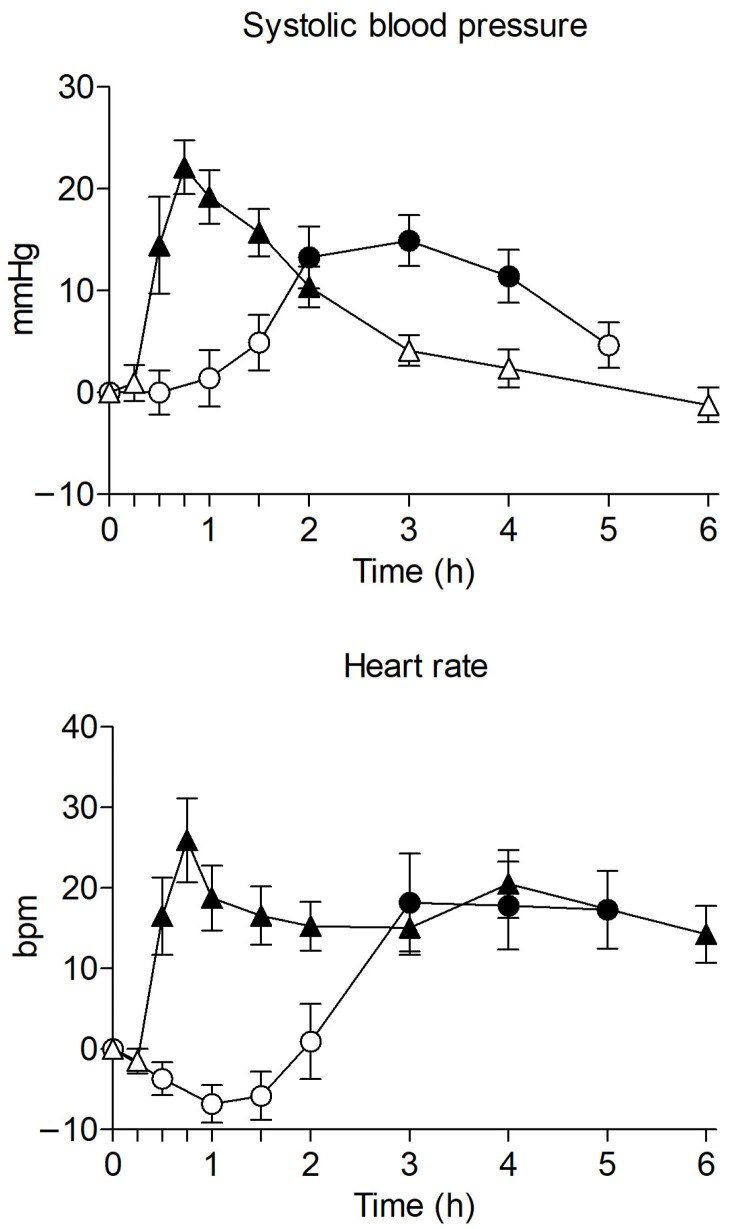
Time course of systolic blood pressure and heart rate after clephedrone (○, 75–100 mg, n = 8, observational study) and methylone (Δ, 150–200 mg, n = 12, clinical trial) oral administration. Results are shown as mean and standard error. Filled symbols represent significant differences with baseline (Dunnett’s test).

**Figure 3 ijms-27-00089-f003:**
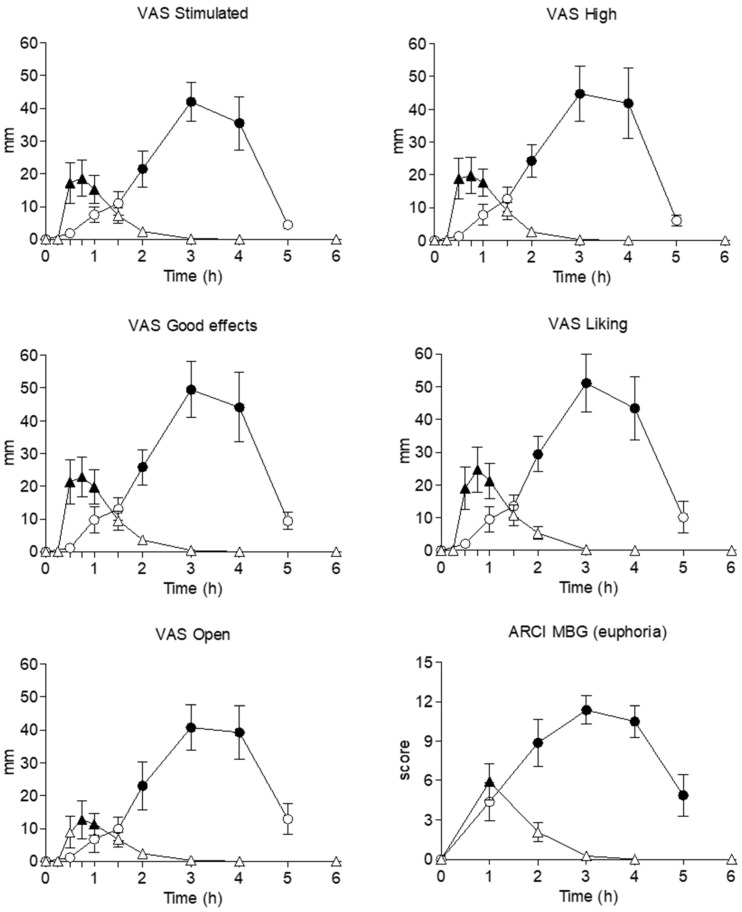
Time course of subjective effects after clephedrone (○, 75–100 mg, n = 8, observational study) and methylone (Δ, 150–200 mg, n = 12, clinical trial) oral administration. Results are shown as mean and standard error. Filled symbols represent significant differences with baseline (Dunnett’s test).

**Figure 4 ijms-27-00089-f004:**
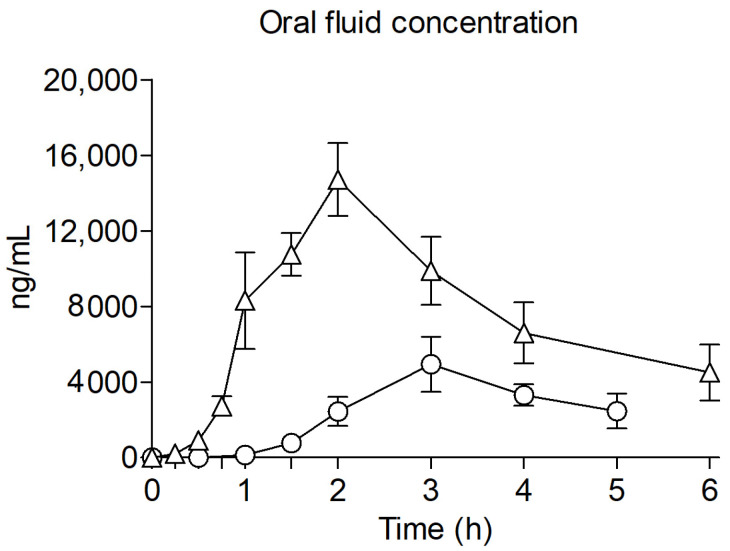
Time course of oral fluid concentration of clephedrone (○, 75–100 mg, n = 8, observational study) and methylone (Δ, 150–200 mg, n = 12, clinical trial) after oral administration. Results are shown as mean and standard error.

**Figure 5 ijms-27-00089-f005:**
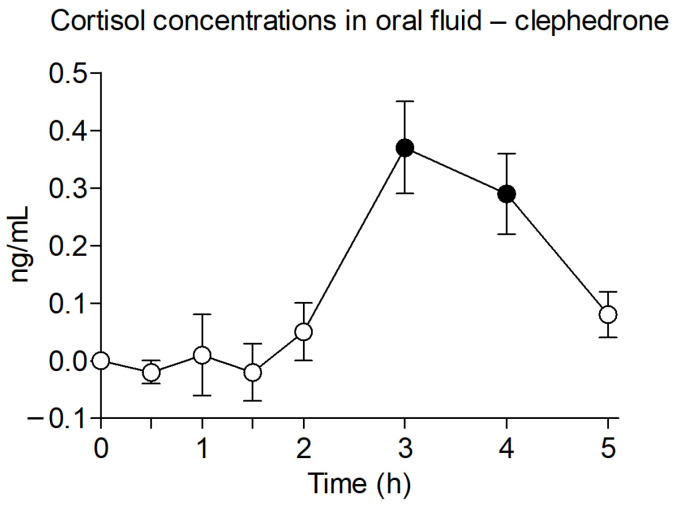
Cortisol concentration in oral fluid after administration of clephedrone (○ 75–100 mg, n = 7, observational study). Results are shown as mean and standard error. Filled symbols represent significant differences with baseline (Dunnett’s test).

**Table 1 ijms-27-00089-t001:** Summary of the statistically significant results on physiological and subjective effects of clephedrone (n = 8, observational study) and methylone (n = 12, clinical trial). Only variables with any reported effect are presented.

		Clephedrone		Methylone	
	Parameters	Mean ± SD/Median (Range)	Dunnett’s Test	Mean ± SD/Median (Range)	Dunnett’s Test
SBP	E_max_ (mmHg)	16.81 ± 13.11	F, G, h	27.04 ± 10.72	B, C, D, E, F
	AUC_0-5/6 h_ (mmHg·h)	41.77 ± 13.86		38.70 ± 26.84	
	T_max_ (h)	3.0 (0.5–4.0)		0.75 (0.5–1.5)	
DBP	E_max_ (mmHg)	13.63 ± 11.74	f, G, H	3.75 ± 11.52	D, E
	AUC_0-5/6 h_ (mmHg·h)	32.64 ± 22.06		00.58 ± 24.15	
	T_max_ (h)	3.0 (0.5–3.0)		1.25 (0.25–6.0)	
HR	E_max_ (bpm)	17.19 ± 25.17	G, H, I	28.75 ± 16.09	B, C, D, E, F, G, H, J
	AUC_0-5/6 h_ (bpm·h)	37.20 ± 40.72		97.07 ± 69.24	
	T_max_ (h)	2.5 (1.0–5.0)		0.75 (0.5–4.0)	
T	E_max_ (°C)	0.63 ± 0.35	B, D, E, F	0.23 ± 0.34	NS
	AUC_0-5/6 h_ (°C·h)	1.25 ± 1.05		0.88 ± 1.34	
	T_max_ (h)	1.5 (0.5–3.0)		1.5 (0.25–6.0)	
VAS Intensity	E_max_ (mm)	49.13 ± 17.18	f, G, H	25.17 ± 18.95	B, C, D
	AUC_0-5/6 h_ (mm·h)	112.03 ± 34.05		20.30 ± 18.35	
	T_max_ (h)	3.0 (2.0–4.0)		0.5 (0.0–1.0)	
VAS Stimulated	E_max_ (mm)	48.75 ± 14.47	f, G, H	25.17 ± 18.95	B, C, D
	AUC_0-5/6 h_ (mm·h)	104.81 ± 29.68		20.30 ± 18.35	
	T_max_ (h)	3.0 (2.0–4.0)		0.5 (0.0–1.0)	
VAS High	E_max_ (mm)	54.25 ± 20.19	f, G, H	26.67 ± 18.12	B, C, D
	AUC_0-5/6 h_ (mm·h)	118.63 ± 44.24		22.85 ± 18.54	
	T_max_ (h)	3.0 (2.0–4.0)		0.75 (0.0–1.0)	
VAS Good effects	E_max_ (mm)	58.50 ± 20.46	F, G, H	29.75 ± 21.60	B, C, D
	AUC_0-5/6 h_ (mm·h)	129.66 ± 50.22		26.38 ± 22.79	
	T_max_ (h)	3.0 (2.0–4.0)		0.63 (0.0–1.0)	
VAS Bad effects	E_max_ (mm)	0.38 ± 0.74	NS	0.17 ± 0.58	NS
	AUC_0-5/6 h_ (mm·h)	0.38 ± 0.74		0.06 ± 0.22	
	Tmax (h)	0.0 (0.0–3.0)		0.0 (0.0–0.5)	
VAS Liking	E_max_ (mm)	59.00 ± 18.56	F, G, H	30.33 ± 23.26	B, C, D
	AUC_0-5/6 h_ (mm·h)	134.09 ± 48.13		28.45 ± 24.43	
	T_max_ (h)	3.0 (2.0–4.0)		0.75 (0.0–2.0)	
VAS Changes in distances	E_max_ (mm)	1.88 ± 4.55	NS	1.58 ± 5.48	NS
	AUC_0-5/6 h_ (mm·h)	1.53 ± 3.76		0.72 ± 2.49	
	T_max_ (h)	0.0 (0.0–2.0)		0.0 (0.0–1.0)	
VAS Changes in colors	E_max_ (mm)	1.13 ± 2.80	NS	0.00 ± 0.00	NS
	AUC_0-5/6 h_ (mm·h)	1.75 ± 4.75		0.00 ± 0.00	
	T_max_ (h)	0.0 (0.0–3.0)		0.0 (0.0–0.0)	
VAS Changes in lights	E_max_ (mm)	4.75 ± 9.79	NS	0.00 ± 0.00	NS
	AUC_0-5/6 h_ (mm·h)	6.41 ± 14.74		0.00 ± 0.00	
	T_max_ (h)	0.0 (0.0–4.0)		0.0 (0.0–0.0)	
VAS Changes in hearing	E_max_ (mm)	8.75 ± 16.24	NS	0.42 ± 1.44	NS
	AUC_0-5/6 h_ (mm·h)	12.44 ± 23.03		0.10 ± 0.36	
	T_max_ (h)	0.0 (0.0–4.0)		0.0 (0.0–0.5)	
VAS Drowsiness	E_max_ (mm)	14.25 ± 22.86	NS	10.17 ± 16.16	NS
	AUC_0-5/6 h_ (mm·h)	13.25 ± 20.39		19.98 ± 33.41	
	T_max_ (h)	1.5 (0.0–5.0)		1.0 (0.0–6.0)	
VAS Concentration	E_max_ (mm)	22.13 ± 16.65	G, h	2.50 ± 4.21	NS
	AUC_0-5/6 h_ (mm·h)	45.34 ± 33.03		1.95 ± 3.58	
	T_max_ (h)	3.0 (0.0–4.0)		0.0 (0.0–2.0)	
VAS Dizziness	E_max_ (mm)	1.13 ± 1.81	NS	3.08 ± 5.33	NS
	AUC_0-5/6 h_ (mm·h)	0.97 ± 1.49		3.31 ± 6.88	
	T_max_ (h)	0.0 (0.0–5.0)		0.0 (0.0–1.5)	
VAS Confusion	E_max_ (mm)	3.00 ± 3.82	NS	1.33 ± 2.57	NS
	AUC_0-5/6 h_ (mm·h)	3.72 ± 5.72		1.46 ± 3.43	
	T_max_ (h)	0.5 (0.0–2.0)		0.0 (0.0–1.5)	
VAS Different or changed body feeling	E_max_ (mm)	39.63 ± 22.15	G, H	12.25 ± 10.68	b, C, d
	AUC_0-5/6 h_ (mm·h)	87.38 ± 52.74		7.96 ± 8.16	
	T_max_ (h)	3.0 (1.0–4.0)		0.5 (0.0–1.5)	
VAS Unreal body feeling	E_max_ (mm)	2.13 ± 4.16	NS	0.00 ± 0.00	NS
	AUC_0-5/6 h_ (mm·h)	3.13 ± 6.75		0.00 ± 0.00	
	T_max_ (h)	0.0 (0.0–4.0)		0.0 (0.0–0.0)	
VAS Different surroundings	E_max_ (mm)	13.75 ± 23.50	NS	0.33 ± 1.15	NS
	AUC_0-5/6 h_ (mm·h)	25.56 ± 52.75		0.17 ± 0.58	
	T_max_ (h)	0.0 (0.0–4.0)		0.0 (0.0–1.5)	
VAS Unreal surroundings	E_max_ (mm)	1.50 ± 4.24	NS	0.00 ± 0.00	NS
	AUC_0-5/6 h_ (mm·h)	1.50 ± 4.24		0.00 ± 0.00	
	T_max_ (h)	0.0 (0.0–4.0)		0.0 (0.0–0.0)	
VAS Open	E_max_ (mm)	49.25 ± 18.78	F, G, H	15.67 ± 19.21	C, D
	AUC_0-5/6 h_ (mm·h)	112.56 ± 51.16		15.04 ± 17.47	
	T_max_ (h)	3.0 (2.0–4.0)		0.75 (0.0–1.5)	
VAS Trust	E_max_ (mm)	53.75 ± 20.62	f, G, H, i	15.92 ± 20.07	C, d
	AUC_0-5/6 h_ (mm·h)	126.91 ± 60.79		15.08 ± 18.74	
	T_max_ (h)	3.0 (2.0–4.0)		0.75 (0.0–1.0)	
VAS Feeling close to others	E_max_ (mm)	53.38 ± 13.79	F, G, H	15.92 ± 19.97	C, D
	AUC_0-5/6 h_ (mm·h)	116.56 ± 42.59		14.20 ± 16.71	
	T_max_ (h)	3.0 (2.0–4.0)		0.75 (0.0–1.5)	
VAS I want to be with other people	E_max_ (mm)	34.63 ± 28.16	h	17.42 ± 20.62	C, d
	AUC_0-5/6 h_ (mm·h)	62.94 ± 49.65		16.94 ± 19.79	
	T_max_ (h)	3.0 (2.0–4.0)		0.75 (0.0–1.5)	
VAS I want to hug someone	E_max_ (mm)	37.63 ± 28.01	G, H	15.33 ± 21.76	c, d
	AUC_0-5/6 h_ (mm·h)	80.38 ± 74.45		14.27 ± 19.51	
	T_max_ (h)	3.0 (0.0–4.0)		0.75 (0.0–1.5)	
VAS Sexual desire	E_max_ (mm)	20.63 ± 19.60	G, h	1.08 ± 2.07	NS
	AUC_0-5/6 h_ (mm·h)	44.69 ± 48.58		5.42 ± 10.33	
	T_max_ (h)	3.0 (0.0–4.0)		0.0 (0.0–1.0)	
VAS Sexual arousal	E_max_ (mm)	20.13 ± 18.70	G, h	0.50 ± 1.45	NS
	AUC_0-5/6 h_ (mm·h)	44.44 ± 49.02		2.50 ± 7.23	
	T_max_ (h)	3.0 (0.0–4.0)		0.0 (0.0–1.0)	
ARCI PCAG	E_max_ (points)	0.88 ± 4.19	d, F, G, H, i	2.17 ± 3.49	NS
	AUC_0-5/6 h_ (points·h)	–3.31 ± 5.44		5.46 ± 8.86	
	T_max_ (h)	2.0 (1.0–5.0)		1.0 (0.0–4.0)	
ARCI MBG	E_max_ (points)	12.13 ± 2.03	F, G, H, i	5.92 ± 4.68	D
	AUC_0-5/6 h_ (points·h)	37.56 ± 11.83		8.25 ± 6.89	
	T_max_ (h)	2.5 (2.0–3.0)		1.0 (0.0–1.0)	
ARCI LSD	E_max_ (points)	1.25 ± 2.19	D, F, G, H, I	–0.25 ± 1.29	NS
	AUC_0-5/6 h_ (points·h)	1.81 ± 4.39		–0.96 ± 1.81	
	T_max_ (h)	2.0 (1.0–4.0)		1.0 (0.0–6.0)	
ARCI BG	E_max_ (points)	4.13 ± 0.99	D, I	1.50 ± 1.09	D
	AUC_0-5/6 h_ (points·h)	10.56 ± 5.30		0.83 ± 1.81	
	T_max_ (h)	2.5 (2.0–4.0)		1.0 (0.0–1.0)	
ARCI A	E_max_ (points)	5.88 ± 0.99	D, F, G, H, I	2.92 ± 1.73	D, F
	AUC_0-5/6 h_ (points·h)	18.75 ± 3.54		4.33 ± 2.77	
	T_max_ (h)	2.0 (2.0–3.0)		1.0 (0.0–1.0)	
VESSPA S	E_max_ (points)	0.90 ± 0.73	i	0.56 ± 0.59	NS
	AUC_0-5/6 h_ (points·h)	1.69 ± 1.43		1.39 ± 1.96	
	T_max_ (h)	2.5 (0.0–5.0)		1.0 (0.0–6.0)	
VESSPA ANX	E_max_ (points)	1.33 ± 0.45	G, H, I	0.92 ± 0.51	D, F, G, h
	AUC_0-5/6 h_ (points·h)	3.01 ± 0.95		2.51 ± 2.22	
	T_max_ (h)	3.0 (2.0–4.0)		1.0 (0.0–3.0)	
VESSPA CP	E_max_ (points)	0.02 ± 0.06	NS	0.00 ± 0.00	NS
	AUC_0-5/6 h_ (points·h)	0.06 ± 0.18		0.00 ± 0.00	
	T_max_ (h)	0.0 (0.0–1.0)		0.0 (0.0–0.0)	
VESSPA SOC	E_max_ (points)	2.85 ± 0.47	F, G, H	1.24 ± 1.56	D
	AUC_0-5/6 h_ (points·h)	7.74 ± 3.00		1.63 ± 2.13	
	T_max_ (h)	3.0 (1.0–4.0)		1.0 (0.0–2.0)	
VESSPA-ACT	E_max_ (points)	2.52 ± 0.46	F, G, H	0.82 ± 0.84	D
	AUC_0-5/6 h_ (points·h)	5.93 ± 2.65		1.02 ± 1.08	
	T_max_ (h)	3.0 (2.0–4.0)		1.0 (0.0–2.0)	
VESSPA-PS	E_max_ (points)	0.38 ± 0.50	f, g	0.06 ± 0.11	NS
	AUC_0-5/6 h_ (points·h)	1.15 ± 2.02		0.10 ± 0.24	
	T_max_ (h)	2.0 (0.0–3.0)		0.0 (0.0–1.0)	

Abbreviations: systolic blood pressure (SBP), diastolic blood pressure (DBP), heart rate (HR), visual analog scale (VAS), Addiction Research Center Inventory (ARCI), ARCI PCAG (sedation), MBG (euphoria), LSD (dysphoria), BG (intellectual efficiency), and A (amphetamine-like effects), Evaluation of Subjective Effects of Substances with Abuse Potential questionnaire (VESSPA), VESSPA S (sedation), VESSPA ANX (psychosomatic anxiety), VESSPA CP (changes in perception), VESSPA SOC (pleasure and sociability), VESSPA ACT (activity and energy), and VESSPA PS (psychotic symptoms). E_max_ and AUC are expressed as mean ± standard deviation, and T_max_ as median (range). Statistical differences for Dunnett’s test between clephedrone or methylone and their respective baseline are presented as “a” *p* < 0.05, “A” *p* < 0.01 (times 0–0.25 h), “b” *p* < 0.05, “B” *p* < 0.01 (times 0–0.5 h), “c” *p* < 0.05, “C” *p* < 0.01 (times 0–0.75 h), “d” *p* < 0.05, “D” *p* < 0.01 (times 0–1 h), “e” *p* < 0.05, “E” *p* < 0.01 (times 0–1.5 h), “f” *p* < 0.05, “F” p < 0.01 (times 0–2 h), “g” *p* < 0.05, “G” (times 0–3 h), “h” *p* < 0.05, “H” (times 0–4 h), “i” *p* < 0.05, “I” (times 0–5 h), and “j” *p* < 0.05, “J” (times 0–6 h).

## Data Availability

Data could be shared with other investigators after evaluating the purpose of the study.

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
