# Peer review of "Oral Fluid Concentrations and Pharmacological Effects of Clephedrone and Methylone in Humans"

_ijms, 2025, doi:10.3390/ijms27010089_

Round 1

Reviewer 1 Report

Comments and Suggestions for Authors

The paper entitled “Oral fluid concentrations and pharmacological effects of clephedrone and methylone in humans” appropriately aims to characterize the pharmacokinetic profile of clephedrone and its acute subjective effects in a naturalistic setting. This experimental approach confers ecological validity, allowing the observations and findings to be generalized to real-world conditions.

This is a highly relevant topic, offering information of considerable value to the field of substance use disorders, particularly in light of the emerging challenges posed by the new psychoactive substances currently in use.

However, I found it difficult to reconcile the two experimental designs presented. One study is described as “non-controlled, prospective, observational, and naturalistic,” whereas the other is a “randomized, double-blind, placebo-controlled, crossover clinical trial.” These designs do not appear directly comparable. Although I understand the authors’ rationale for drawing parallels between them, several methodological factors—such as differences in participants’ sex, the experimental setting in general (the dose, the self-administration and self-collection of the drug)  —are substantial. As the authors note, it is indeed important to compare first- and second-generation synthetic cathinones in terms of their pharmacological actions and temporal profiles, that is, pharmacokinetic variables such as the time to reach peak plasma concentration (Tmax) and the duration of effects relative to elimination half-life. However, it is still hard to understand the intention of comparing as if the studies were done simultaneously. I would appreciate clarification on what the authors mean by “contextual comparison.” The manuscript states that the two studies share “comparable methodological conditions,” yet the methodologies differ markedly.

I recommend that the manuscript be more tightly focused on the results of clephedrone (CMC), allowing the narrative to follow a single, coherent experimental framework when discussing its physical and psychological effects. The methylone findings (unpublished) could then be incorporated as a historical reference group, particularly since methylone is more represented in the existing literature. This structure would likely strengthen the manuscript’s internal consistency and interpretive clarity.

In this context, I am uncertain whether it is appropriate to present the results of both studies in a single figure, given that they are not being compared directly, but rather each dataset is analyzed relative to its own baseline. It would be helpful for the authors to address this concern. They might consider a more conventional approach to presenting the data—if feasible, by separating the studies. Such a revision would also make the manuscript less burdensome to read, as it would avoid the need to repeatedly revisit the details of each study for each result.

The introduction, discussion, and overall manuscript are relevant and well written; however, the methodology and results sections are somewhat challenging to read, likely because there is no alternative way to describe them more simply.

Minor errors:

-line 202: a floating “were recruited.” at the beginning of the line.

Author Response

Thank you very much for taking the time to review this manuscript. Please find detailed responses below and the corresponding revisions/corrections highlighted in the re-submitted files. We think that the responses will improve the Questions for General Evaluations done by the reviewer.

Comment 1: This is a highly relevant topic, offering information of considerable value to the field of substance use disorders, particularly in light of the emerging challenges posed by the new psychoactive substances currently in use.

However, I found it difficult to reconcile the two experimental designs presented. One study is described as “non-controlled, prospective, observational, and naturalistic,” whereas the other is a “randomized, double-blind, placebo-controlled, crossover clinical trial.” These designs do not appear directly comparable. Although I understand the authors’ rationale for drawing parallels between them, several methodological factors—such as differences in participants’ sex, the experimental setting in general (the dose, the self-administration and self-collection of the drug)  —are substantial. As the authors note, it is indeed important to compare first- and second-generation synthetic cathinones in terms of their pharmacological actions and temporal profiles, that is, pharmacokinetic variables such as the time to reach peak plasma concentration (Tmax) and the duration of effects relative to elimination half-life. However, it is still hard to understand the intention of comparing as if the studies were done simultaneously.

Response 1:

Thank you for your constructive comment regarding the two studies included in the manuscript. You are correct that, in general terms, the reconciliation of the two experimental designs is a key issue because the designs are different. This response answers related comments (Comments 1, 4 and 5).

We agree that two different designs seem “a priori” to be a significant challenge for integrating the results. However, the very nature of the differences obtained is a strong argument in favor of including both the methylone and clephedrone studies in the paper.

Our main objective was to include both studies in the same manuscript to emphasize the relevant differences in the onset and time-course of effects and pharmacokinetics between two distinct synthetic cathinones. Presenting them together provides a unique opportunity to demonstrate these differences simultaneously, even though they resulted from distinct study designs. We want to explain that the same methods were used for evaluating physiological effects, the same questionnaires were used to measure subjective effects, and the same group of investigators carried out both studies.

In our opinion, concentrating the manuscript solely on clephedrone would prevent us from presenting the important finding that different synthetic cathinones can produce relevant differences in the  onset-time-course of pharmacological effects and pharmacokinetics. While the main pharmacological effects on subjective measures, vital constants, and cortisol are in the same direction for both substances—thus demonstrating group- or class-effects—the time-course is distinct. Given the absence of comparative studies between synthetic cathinones, separating the methylone data from the manuscript would significantly reduce the opportunity to communicate these crucial differences.

We believe that figures that display the data for both substances side-by-side are an excellent way to inform readers about the relevant differences in the onset and time-course of effects and pharmacokinetics. Separate figures or only presenting data on clephedrone will reduce the impact of the paper, whose main purpose is to demonstrate these contrasting time-course patterns.

We propose to include both studies but explaining the differences in the designs much more clearly within the text, including in the legends of the Tables and Figures, and in the limitations section.

The modifications are:

End of introduction: “The inclusion of both studies in the same manuscript—despite their very different designs (an observational study and a randomized clinical trial)—will permit the comparison of the onset-time-course of pharmacological effects, and pharmacokinetics of two synthetic cathinones from different generations.”

Table 1. “Summary of the statistically significant results on physiological and subjective effects of clephedrone (n=8, observational study) and methylone (n= 12, clinical trial). Only variables with any reported effect are presented”.

Figures 2, 3 and 4. Legends: We added in the legends the design of the studies “(clephedrone (○, 75–100 mg, n=8, observational study) and methylone (Δ, 150–200 mg, n= 12, clinical trial)”. Figure 5: “(○, 75–100 mg, n=7, observational study)”.

Limitations: “Regarding limitations, the inclusion of two studies with distinct designs (an observational study and a randomized clinical trial) is a relevant point that must be acknowledged. However, given the highly interesting results observed following clephedrone self-administration, we believe that a direct comparison with methylone provides original and previously unobserved differences. Two different synthetic cathinones can induce similar overall pharmacological effects, yet present markedly different profiles in terms of time of onset and time-course. This finding is of significant clinical relevance. Therefore, the inclusion of both studies is essential for understanding that molecules within the same class can exhibit distinct pharmacokinetic profiles.”

Comment 2: I would appreciate clarification on what the authors mean by “contextual comparison.”

Response 2: In reference to the “contextual comparison.”. We modified the sentence in two parts of the text as: “For comparison”

Comment 3: The manuscript states that the two studies share “comparable methodological conditions,” yet the methodologies differ markedly.

Response 3: In reference to the use of “comparable methodological conditions,” yet the methodologies differ markedly. We modified the sentence explaining that we refer to methods of evaluation of pharmacological effects, as subjective effects questionnaires. The text has been modified in abstract as: “comparable methods to evaluate effects”

Comment 4: I recommend that the manuscript be more tightly focused on the results of clephedrone (CMC), allowing the narrative to follow a single, coherent experimental framework when discussing its physical and psychological effects. The methylone findings (unpublished) could then be incorporated as a historical reference group, particularly since methylone is more represented in the existing literature. This structure would likely strengthen the manuscript’s internal consistency and interpretive clarity.

Response 4. The previous comment pertains to the inclusion of both studies, and we refer the reviewer to our Global Response 1 for full justification.

We acknowledge the complexity of incorporating the methylone findings (currently unpublished) to serve as a historical reference group. Specifically, if the methylone data is included, we are obligated to describe that study with the same level of methodological detail and rigor as the main clephedrone study, to maintain consistency and allow the results to be properly contextualized and interpreted.

Comment 5: In this context, I am uncertain whether it is appropriate to present the results of both studies in a single figure, given that they are not being compared directly, but rather each dataset is analyzed relative to its own baseline. It would be helpful for the authors to address this concern. They might consider a more conventional approach to presenting the data—if feasible, by separating the studies. Such a revision would also make the manuscript less burdensome to read, as it would avoid the need to repeatedly revisit the details of each study for each result.

Response 5: The previous comment pertains to the inclusion of both studies, and we refer the reviewer to our Global Response 1 for full justification. To avoid a direct statistical comparison, each dataset was analyzed relative to its own baseline. This limitation es described in the corresponding section.

Comment 6: The introduction, discussion, and overall manuscript are relevant and well written; however, the methodology and results sections are somewhat challenging to read, likely because there is no alternative way to describe them more simply.

Response 6: Thank you for your comment. We agree that including both studies is very difficult to simplify the methods and results sections. We did not change this section.

Comment 7: Minor errors:

-line 202: a floating “were recruited.” at the beginning of the line.

Response 7. Agree. Modified

Reviewer 2 Report

Comments and Suggestions for Authors

Dear Authors, 

Manuscript entitled "Oral fluid concentrations and pharmacological effects of clephedrone and methylone in humans" is well drafted. 

Please update/ justify the points below.

  1. Paragraph or lines 575 to 579 are presented (copied) a second time in lines 580 to 584.
  2. In section 2.6, for clephedrone, what is the rationale for collecting oral fluid samples only till 5 hours or stopping the sample collection at 5 hours?
  3. As mentioned in the manuscript, this study has significant limitations, not limited to design, small sample size, etc., which affect the findings. Please try to provide detailed justifications. 
  4. Please elaborate on the Tmax (h) values of VAS bad effects, change in distances, in colors, in lights, in hearing, dizziness, body feeling, surrounding etc., observations or readings. 

Author Response

Reviewer 2. Point-by-point responses

Thank you very much for taking the time to review this manuscript. Please find detailed responses below and the corresponding revisions/corrections highlighted in the re-submitted files. We think that the responses will improve the Questions for General Evaluations done by the reviewer.

Comment 1: Paragraph or lines 575 to 579 are presented (copied) a second time in lines 580 to 584.

Response 1: Deleted copied paragraph in lines 580-584, in addition sentences modified.

Comment 2: In section 2.6, for clephedrone, what is the rationale for collecting oral fluid samples only till 5 hours or stopping the sample collection at 5 hours?

Response 2:

Since the study utilized an observational design, we adjusted the schedule to perform the study in the afternoon and included a telephone control 24 hours later. The collection of oral fluid samples for clephedrone was limited to five hours based on the study protocol and the assumption of rapid elimination. This assumption was derived from the effects and pharmacokinetics previously observed by our group with structurally similar cathinones (mephedrone, methylone). We initially anticipated that this five-hour period would be sufficient to characterize the drug profile. However, analysis of the results suggests that an  additional sample at seven hours would have been beneficial for accurately detecting the very low terminal concentrations in the oral fluid. This aspect is discussed in the Limitations section of the manuscript as:

“Due to the delayed onset of clephedrone effects, additional assessments after 5 h would have allowed to define the time course of its pharmacological effects and oral fluid con-centrations until returning to baseline.”

Comment 3: As mentioned in the manuscript, this study has significant limitations, not limited to design, small sample size, etc., which affect the findings. Please try to provide detailed justifications.

Response 3: We have modified the section to provide more detailed justifications.

Regarding limitations, the inclusion of two studies with distinct designs (an observational study and a randomized clinical trial) is a relevant point that must be acknowledged. However, given the highly interesting results observed following clephedrone self-administration, we believe that a direct comparison with methylone provides original and previously unobserved differences. Two different synthetic cathinones can induce similar overall pharmacological effects, yet present markedly different profiles in terms of time of onset and time-course. This finding is of significant clinical relevance. Therefore, the inclusion of both studies is essential for understanding that molecules within the same class can exhibit distinct pharmacokinetic profiles

For the controlled methylone study, the main limitations were the exclusion of female participants, the limited number of subjects, and the restricted range of doses included. Although the sample size was sufficient to achieve statistically significant results regarding the time-course, a larger sample size might have revealed differences between the two doses, which were ultimately aggregated due to the lack of observed effect differences. Finally, cortisol concentration data were unavailable for comparison.

In particular, the study about clephedrone had limitations inherent to its design as an observational-naturalistic. The sample size used in the naturalistic study was insufficient to explore sex influence in the acute effects and/or pharmacokinetics of clephedrone. Moreover, differences in effects based on the doses could not be detected, nor could a dose-response relationship be established. The study was open label, so it is susceptible to an expectancy bias. But as explained in other studies the observational approach and the results presented in the present paper can produce adequate results when controlled clinical trials cannot be performed [14,16]. Due to the delayed onset of clephedrone effects, additional assessments after 5 h would have allowed to define the time course of its pharmacological effects and oral fluid concentrations until returning to baseline. Another limitation, as commented previously, refers that a comparative analysis between both cathinones could not be performed due to the difference in study design (observational study and clinical trial) that could have influenced on the subjective effects reported by the participants, in general open studies produce more subjective effects than controlled double-blind studies.

Comment 4: Please elaborate on the Tmax (h) values of VAS bad effects, change in distances, in colors, in lights, in hearing, dizziness, body feeling, surrounding etc., observations or readings.

Response 4: Thank you for your comment.  Tmax (time of maximum effect/concentration) is considered non-parametric data and median and range statistics are recommended. In our study, particularly for variables like the Visual Analog Scale (VAS) scores, the data distribution was heavily skewed, containing many zero values.

As an example, in the VAS bad effects of the observational study (see Table 1), six participants scored 0 mm and two participants scored 2 and 1 mm at 1h and 3h, respectively. The dataset of Tmax is 0h,1h, 0h, 0h, 0h, 0h, 3h, and 0h (sorted dataset: 0, 0, 0, 0, 0, 0, 1, 3), which further illustrates the prevalence of zero values.  For these data, the median is 0h and range 0-3h.             

Round 2

Reviewer 1 Report

Comments and Suggestions for Authors

In the attached file, I point out minor errors that detract from the manuscript's seriousness. These should be carefully revised.

Author Response

Reviewer 1. Round 2.

Thank you very much for taking the time to review this manuscript and your suggested modifications to minor errors in the text.

We corrected the errors in the new R2 version of the manuscript.

Please find the detailed responses below and the corresponding corrections are highlighted/deleted in the re-submitted files (R2)

Comment 1. Lines 54, 56, 59-60, 86, 93, 98, 117, 128, 134-35, 175-176, 196, 201, 202, 543, 545, 568, 592, 613 and 630.

Response 1. Thank you for pointing this out. Minor errors have been modified as suggested in each line/s. The text has been highlighted/deleted, as needed (R2)

Comment 2. The description of subjective effect measurements is not consistent, as the items of the VESSPA-SSE questionnaire are not specified (36 items?)

Response 2. The description of VESSPA has been modified as: “The questionnaire includes 36-items and consist of six subscales that”
